# Third-Generation Snacks Manufactured from Andean Tubers and Tuberous Root Flours: Microwave Expansion Kinetics and Characterization

**DOI:** 10.3390/foods12112168

**Published:** 2023-05-27

**Authors:** Liliana Acurio, Diego Salazar, Purificación García-Segovia, Javier Martínez-Monzó, Marta Igual

**Affiliations:** 1G+ BioFood and Engineering Research Group, Department of Science and Engineering in Food and Biotechnology, Technical University of Ambato, Av. Los Chasquis and Río Payamino, Ambato 180150, Ecuador; dm.salazar@uta.edu.ec; 2Food Technology Department, Universitat Politècnica de València, Camino de Vera s/n, 46021 Valencia, Spain; pugarse@tal.upv.es (P.G.-S.);

**Keywords:** extrusion, purple sweet potato, mashua, tropaeolum tuberosum, oxalis tuberosa

## Abstract

Andean tubers and tuberous roots have nutritional and medicinal properties transferred through ancestral generations. In this study, we aim to promote cultivation and consumption by developing a snack based on these crops. Corn grits were thoroughly mixed with sweet potato, mashua, and three varieties of oca flour (white, yellow, and red) in an 80:20 ratio, and a single-screw laboratory extruder was utilized to produce third-generation (3G) dried pellets. Microwave expansion was studied, and the dried 3G pellets and expanded snacks were characterized. The microwave expansion curves of the dried 3G pellets were adjusted to the Page, logarithmic, and Midilli–Kucuk models. During the characterization, the influence of the raw material composition was observed in sectional expansion, water content, water activity, water absorption, water solubility, swelling, optical and textural properties, and bioactive compounds. According to global color variation (mixture vs. expanded and dried vs. expanded) and bioactive compound analysis, the mashua suffered little chemical change or nutritional loss during the process. The extrusion process was shown to be an ideal method for manufacturing snacks from Andean tuber flours.

## 1. Introduction

The Andean region is an area with extraordinary diversity of tubers and tuberous roots due to geographical and climatological conditions [1]. The foremost critical prerequisites are elevation (500 to 5000 masl), daytime temperature (around 23 °C), yearly precipitation (752 mm/year), and zones with high sun-oriented radiation (18.8 in clear sky conditions) [2]. The total production in this region in 2020 of potatoes, cassava, sweet potatoes, and roots and tubers not specified elsewhere was approximately 14.02, 3.41, 0.63, and 0.58 megatons, respectively [3].

Due to their nutritional and medicinal properties, indigenous Andeans consume most of these roots and tubers (Figure 1) [4]. These properties have been transferred through generations in ancestral knowledge and are directly related to the types of chemical components present. One of the most commonly consumed tuberous roots is “camote morado” or purple sweet potato (*Ipomoea batatas* (L.) Lam.). This crop is also known as purple potatoes, and its color reveals the presence of anthocyanins. Many studies elucidate this root’s high total phenol and anthocyanin content and antioxidant activity [5]. Purple sweet potato anthocyanins have high heat resistance [6]; therefore, from a technological viewpoint, they are an excellent raw material for preparing different foods [7].

Mashua (*Tropaeolum tuberosum* Ruiz and Pavón) is a crop with a short cultivation cycle (5–6 months) compared with other root and tuber crops [8]. Tubers are ovoids, and the skin is usually yellow or occasionally purplish or red [9]. Leaves and blossoms can be eaten, but the tuber is the most commonly consumed part due to its flavor and nutritional value. It contains protein (1.5 g/100 g), fiber (0.9 g/100 g), and carotenes (1 μg/g_db_), and its vitamin C content is four times higher than that of the potato (77 mg in 100 g of fresh edible matter) [10,11].

Another widely consumed tuber is Oca (*Oxalis tuberosa* Molina), which produces ovoid tubers (7 to 11 cm in length) whose colors vary from green to red. The most common varieties are white, yellow, and red. This crop grows in the highlands of Bolivia, Ecuador, and Peru, but is also found in the rest of the Andes. Nutritionally, tubers are an essential source of carbohydrates (starch 42.17%), total sugars (9.68%), vitamins (A, B_1_, B_2_, Niacin, and C), and minerals (calcium, iron, phosphorus, and zinc) [12,13].

Extrusion allows the production of foods that are either directly expanded (second generation product—2G) or indirectly expanded by hot-air heating, baking, frying, or microwave heating (third generation product—3G) [14]. This last type of food, also called pellets, exhibits a stable water content and high bulk density [15,16]. Corn, potato, rice, wheat flour, and starch are usually used to manufacture 3G pellets [17,18,19,20,21].

The use of microwaves presents essential advantages over other traditional methods. For example, from the microbiological viewpoint, some studies declare it to be an efficient treatment for eliminating microorganisms, including pathogens, from food samples [22]. Furthermore, from the viewpoint of the stability of bioactive compounds, there is scientific evidence for the conservation of thermolabile compounds when microwaving is used as an extraction method for these compounds in herbal samples [23].

The microwave expansion process (1000 W/g) has been analyzed in corn extruded 3G pellets using corn flour samples with 25, 30, and 35% (wb) water content [24]. In the study, the pellets with 25% moisture expanded after 40 s of microwave application, whereas above 75 s, all pellets burned due to the loss of water content. According to the sectional and volumetric expansion indices and appearance evolution, the best treatment was a pellet with 25% water content and a microwave drying time of 40 s.

The objectives of this work were (1) to examine the effect of Andean tubers and tuberous root flour on the microwave expansion kinetic, and (2) to explore the effect of these flours on appearance, SEI, and different characteristics such as water content, water activity, bulk density, porosity, hygroscopicity, water absorption index (WAI), water solubility index (WSI), swelling index (SWE), texture and optical properties, and bioactive compounds. Using these roots is expected to add nutritional characteristics to this type of food.

## 2. Materials and Methods

### 2.1. Raw Materials

Sweet potato (*I. batatas* (L.) Lam.), mashua (*T. tuberosum* Ruiz and Pavón), and three varieties of Oca (*O. tuberosa* Molina) (white, yellow, and red) were purchased from a local market (Ambato, Ecuador). Corn grits were purchased from Maicerías Españolas S.L. (València, Spain).

### 2.2. Andean Tubers and Tuberous Roots Used for Flour Manufacturing

The tubers and tuberous roots were peeled and cut into slices (thickness: 2 mm). The slices were pretreated in microwaves (750 W/20 s) and then submerged in water at 4 °C for 20 s. Pretreatment was conducted to avoid enzymatic browning and was defined based on previous studies and preliminary tests [25,26,27]. Subsequently, the slices were dried by convection in a dehydrator model CD 160 (Gander Mountain, Saint Paul, MN, USA) at 65 °C for 8 h. The dehydrated slices were crushed in electric mills model 80393 (Hamilton Beach, Picton, ON, Canada), in 3 intervals of 10 s each to prevent the matrices from increasing in temperature, causing the denaturation of the organic compounds of interest. The flour obtained was preserved in hermetic aluminized bags.

### 2.3. Production of Extruded 3G Pellets

For the control, corn grits were thoroughly mixed with water and adjusted to achieve a 25% wet base (wb) by continuously mixing at medium speed in a mixer model MFQ40303 (Bosch, Gerlingen, Germany). For the samples, corn grits were thoroughly mixed with the Andean tubers and tuberous root flour in an 80:20 ratio (80% corn grits: 20% sample flour), and water was added to adjust to 25% wb by continuously mixing at medium speed in the same mixer. The following samples were obtained: C, control; IbP, purple sweet potato flour; Tt, mashua flour; OtW, Oca white variety flour; OtY, Oca yellow variety flour; OtR, Oca red variety flour.

A single-screw laboratory extruder model KE 19/25 (Brabender, Duisburg, Germany, length-diameter ratio of 25:1) was used to obtain extruded 3G pellets. The extruder was operated at a compression ratio of 3:1 and loaded with prepared samples at a constant dosing speed of 20 rpm. The screw was constantly rotated at 120 rpm, and the temperatures of barrel Section 1, Section 2, Section 3 and Section 4 were set to 30, 60, 100, and 120 °C, respectively. The nozzle diameter was 3 mm. The extruded 3G products were immediately dried at 25 °C for 18 h. The dried samples (dried 3G pellets) were stored in polyethylene bags at 25 °C and used for further analysis.

### 2.4. Microwave Dehydration of Dried 3G Pellets

The expansion process was performed in a microwave model FT339 (Whirlpool Corporation, Benton Harbor, MI, USA) at 1000 W/g. The water content and width and the length of the pellets were analyzed at 10 s intervals (from 10 to 90 s) during microwave drying to evaluate the microwave drying kinetics of the different dried 3G pellets.

#### 2.4.1. Microwave Dehydration Kinetics of Dried 3G Pellets

The microwave dehydration curves, obtained from mean values of three experimental determinations of the water content (x_w_) of the samples, were fitted to the Page (MR = exp (−kt^n^)), logarithmic (MR = a exp (−kt) + c), and Midilli–Kucuk (MR = a exp (−kt^n^) + bt) models, as proposed by Igual et al. [28] for freeze-dried grapefruit. Moisture ratio (MR) represents the amount of moisture remaining in the samples reported in relation to the initial moisture content.

The x_w_ was obtained by vacuum-drying the samples in a vacuum oven (Vaciotem, J.P. Selecta, Barcelona, Spain) at 60 °C for 48 h.

#### 2.4.2. Microwave Expansion Kinetics of Dried 3G Pellet

The SEI evolution of the dried 3G pellet as a function of the microwaving time was determined following the methodology described by Patil et al. [29]. The width and length were measured with a digital caliper.

### 2.5. Characterization of Dried 3G Pellets and Expanded Snacks

#### 2.5.1. Water Content of Mixture, Extruded 3G Pellets, Dried 3G Pellets, and Expanded Snacks, and Water Activity of Dried 3G Pellets

The x_w_ was obtained by vacuum-drying the samples in a vacuum oven (Vaciotem, J.P. Selecta, Barcelona, Spain) at 103 °C for 48 h. The water activity of the dried 3G pellets was determined using a hygrometer AquaLab PRE (Decagon Devices, Inc., Pullman, WA, USA).

#### 2.5.2. Hygroscopicity of Dried 3G Pellets and Expanded Snacks

Hygroscopicity (Hy) was determined following the methods of Cai and Corke [30]. The samples were placed in a Petri dish at 25 °C in an airtight plastic container containing saturated Na_2_SO_4_ solution (81% relative humidity). After seven days, each sample was weighed, and the Hy was expressed as grams of water gained per 100 g_dry solids_.

#### 2.5.3. Bulk Density and Porosity of Expanded Snacks

The bulk density (ρ_b_) was calculated by dividing the mass of the product by volume and expressed as g/m^3^, as described by Gujska and Khan [31]. The diameter and height of the cylinders were measured with an electronic Vernier caliper (Comecta S.A., Barcelona, Spain), and the samples were weighed with a precision scale (±0.001 g) (Mettler Toledo, Greifensee, Switzerland). The density (ρ) of the expanded sacks was determined using a helium pycnometer model AccPyc 1330 (Micromeritics, Norcross, GA, USA). Furthermore, the porosity (ε) was calculated from density (ρ) and bulk density (ρ_b_) as described by García-Segovia et al. [32]. The results were the average of 10 measurements.

#### 2.5.4. Water Absorption Index (WAI), Water Solubility Index (WSI), and Swelling Index (SWE) of Expanded Snacks

These parameters were determined using the method of Singh and Smith [33] and calculated as described by Uribe-Wandurraga et al. [34]. The SWE was measured using the bed volume technique and expressed as millimeters of swollen sample per gram of dry initial sample [35].

#### 2.5.5. Texture Properties of Expanded Snacks

Texture properties were measured using puncture tests in a TA-XT2 Texture Analyzer (Stable Micro Systems Ltd., Godalming, UK) and software (Texture Exponent, version 6.1.12.0). The area under the force-time curve plot was determined as the work done by the puncturing device. The average puncturing force (F_p_), the average specific force of structural ruptures (F_s_), the spatial frequency of structural ruptures (N_sr_), and the crispness work (W_c_) were calculated using the force drop of each peak and the number of peaks (N_0_) [36,37].

#### 2.5.6. Optical Properties of Mixture, Dried 3G Pellets, and Expanded Snacks

These properties were determined with the CIE*L*a*b* color coordinates from the surface reflectance spectra obtained between 400 and 700 nm, considering a standard light source D65 and a standard observer 10 in a spectrophotometer CM-3600d (Minolta, Tokyo, Japan). Hue (h*) and chroma (C*) color attributes were calculated from CIE*L*a*b* color coordinates. The total color difference between the mixture before extrusion and the finished expanded product (ΔE_1_) was calculated for the samples. Furthermore, the total color differences between the dried 3G pellets and the finished expanded product (ΔE_2_) were calculated.

#### 2.5.7. Bioactive Compounds of Expanded Snack

Total carotenoids (TC) were extracted with an acetone/hexane/ethanol solvent mixture following the method of Olives-Barba et al. [38]. The AOAC spectrophotometric reference method [39] was used for TC quantification. The absorbance was measured at a wavelength of 446 nm in a UV-3100PC spectrophotometer (VWR, Leuven, Belgium). The TC content was expressed as mg of β-carotene (Sigma-Aldrich, Steinheim, Germany) per 100 g of sample (mgβ-carotene/100 g). From the TC extract, lycopene (LP) was determined, and the sample absorbance was measured at 503 nm. The LP was calculated as described by Khamis et al. [40], and the content was expressed as mg/100 g of sample.

Total phenols (TP) were determined according to the method described by Igual et al. [41], using the Folin–Ciocalteu method with methanol extraction. The absorbance of the sample at 765 nm was measured in an UV-3100 PC, VWR (Radnor, Philadelphia, PA, USA) and expressed as mg gallic acid/100 g of dry solid sample (mg GAE/100 g_dry solid_).

The antioxidant capacity (AC) was determined using the DPPH method, as previously described by Igual et al. [41]. The samples were mixed with methanol, and 0.1 mL of this extract was mixed with 3.9 mL DPPH (0.030 g/L) and stored for 5 min. The absorbance of the sample was measured at 515 nm and expressed as milligram Trolox equivalents (TE) per 100 g dry solid sample (mg TE/100 g_dry solid_).

### 2.6. Statistical Analysis

Nonlinear regression analyses were conducted for the estimation of the kinetic parameters. Analysis of variance (ANOVA) was applied, with a confidence level of 95% (*p <* 0.05), to evaluate differences among dried 3G pellets and expanded snacks. Tukey’s multi-range test was used to establish the relationships among the studied parameters. Statgraphics Centurion XVII Software, version 17.2.04 (Statgraphics Technologies, Inc., The Plains, VA, USA) was used in the analyses.

## 3. Results and Discussion

### 3.1. Microwaving Dehydration Kinetics of Dried 3G Pellets; Mathematical Modeling

The dehydration curves of the dried 3G pellets were obtained by plotting the moisture ratio vs. time. Figure 2 shows the experimental behavior and the curves adjusted with the Page, logarithmic, and Midilli–Kucuk models. Most samples show a sigmoidal behavior; however, the OtY kinetic curve shows a linear trend. The accuracy of the fit determined for the three models (by R^2^ and root-mean-square error (RMSE)) and the kinetic constants are presented in Table 1. These models coincided well with the experimental data, as seen from the adjusted regression coefficient (R^2^) and the RMSE values. The best fit (higher R^2^) was the Page model for the C, Tt, and OtY samples, whereas the Midilli–Kucuk model adjusts the experimental data of IbP, OtW, and OtR. Therefore, analyzing the two fit parameters (R^2^ and RMSE), the Page model provided the best fit for all the samples analyzed.

For the Page models, the constant k showed a higher value in OtR and C, indicating a higher water diffusion velocity in these samples (higher moisture ratio changes during the period observed) [42]. This model’s constant n showed values higher than the unit, which is associated with a super-diffusion process, and this constant is an empirical coefficient [24]. The constant k, known as the drying constant, provides the most information. Higher values of k are observed in the logarithmic model and lower values in the Midilli–Kucuk model. This trend is typical and has been reported in microwave-dried foods such as apple slices [43], peppermint leaves [44], and white mulberry [45], among others. Furthermore, there is evidence of a positive correlation of k with microwave power level [46] and temperature [47] and an inverse correlation with sample thickness [46].

The modeling of dehydration kinetics provides information that can be related to the food changes during drying [48]. For example, it can be deduced that OtW and Tt are matrices that reach low moisture values in short periods of time, while the OtR, C, IbP and OtY samples present greater resistance to drying (Figure 2). The observed trend is related to the optimal expansion time for each sample. The OtW and Tt dried 3G pellets need less time to expand (50 s), while OtY expands in 60 s and the rest of the samples (IbP and OtR) expand in 70 s (Figure 3 and Figure 4). Although the models are empirical, this analysis shows that these involve parameters that can be related to the changes exerted in the moisture-removal process. Therefore, it has great significance in designing and optimizing the 3G expansion process to produce high-quality snacks.

### 3.2. Sectional Expansion Index (SEI) and Appearance of Dried 3G Pellet as A Function of Microwaving Time

Figure 3 shows the evolution of the dried 3G pellet SEI as a function of the processing time for all samples. At the beginning of the process (10 s), microwave energy heated the matrix progressively; however, the pellets did not show notable changes in SEI values. Then, at 15 s, the Tt, OtW, and OtR pellets expanded, while the other samples (C, IbP, and OtY) expanded at 20 s. The Tt and OtW needed 50 s to expand, whereas the OtW and OtR pellets needed 60 s and IbP needed 70 s. The pellet that required the longest time was the control pellet (90 s) (Figure 4). According to the SEI values and appearance, the best treatment was that of the IbP pellet, with a microwave drying time of 70 s. The samples with the lowest SEI values were OtY and Tt. According to scientific evidence, the interaction of starch and protein in the sample influences gel formation and the increase in the paste viscosity during the expansion process [49,50], ultimately causing a greater expansion level. This would explain the higher SEI values of the IbP pellet, since the raw root contains more starch (40 ± 2%) [25,51] compared to Tt (22 ± 0.3%) and OtY (28 ± 0.4%) [25,52].

### 3.3. Characterization of Dried 3G Pellets and Expanded Snacks

#### 3.3.1. Water Content of the Mixture, Extruded 3G Pellets, Dried 3G Pellets, and Expanded Snacks, and Water Activity of the Dried 3G Pellets

The extrusion process reduced approximately 5.6% of the mixture’s water content due to heating, puffing, and partial drying (Figure 5). In this complex process, water evaporates due to increased temperature and is also involved in the starch gelatinization process, forming new structures of molecular aggregates through hydrogen bonds [53]. The sample with the lowest humidity reduction in the extrusion process was the OtY, with 4.5%. The drying process evidenced a reduction in the water content of around 8.9% in all samples (comparing extruded 3G pellets and dried 3G pellets). All samples showed values after this phase of 11 ± 0.49 g_water_/g_dry mass_, which is a usable quality from the industrial viewpoint because they can be considered foods with stable storage moisture. Finally, the microwave drying stage showed the most considerable decrease in the water content, with approximately 21.5% (comparing mixtures and expanded snacks). Contrary to what was observed in the extrusion phase, the OtY sample had the most significant reduction in water content, considering the three processing phases (extrusion, drying, and microwave drying (expansion)). When the samples were analyzed, OtW presented the highest moisture values in all phases, whereas IbP presented the lowest values in all stages. Analyzing the raw material’s moisture, IbP had a 6.20 ± 0.27% value, whereas the OtW had 16.40 ± 0.18%. Therefore, the influence of the raw material’s water content in the processing phases was evident.

The water activity of the dried 3G pellets showed higher values in the OtY sample (0.37 ± 0.003), a behavior consistent with what was previously discussed in regard to the water content (Table 2). This variety offers a different behavior from dried 3G pellets made with the other samples (OtW and OtR) of the same species (*O. tuberosa* Molina). The sample with the second highest water activity was Tt, with an average value of 0.348 ± 0.003. Finally, the pellet with the lowest water activity was IbP (0.233 ± 0.003). As for the water content, the water activity of the pellets is related to the water activity of the raw material. In this study, mashua (*T. tuberosum* Ruiz and Pavón) had water activity of 0.9894 ± 0.0047, whereas purple sweet potato (*I. batatas* (L.) Lam.) had a lower value (0.9827 ± 0.0034). Therefore, there was an influence of the water activity of the raw material, a property that is also influenced by the molecular features of the root [54].

#### 3.3.2. Bulk Density and Porosity of Expanded Snacks

The bulk density values ranged between 0.15 and 0.23 g/cm^3^. The low values obtained show an advantage from the industrial viewpoint since a low density allows packing a greater quantity of product mass in a smaller volume, reducing distribution and export costs. These values are similar to those reported in corn snacks with a 10% substitution of beetroot by-products, with the same percentage of water content in the mix (25%) at 0.246 ± 0.013 g/cm^3^ [55]. However, porosity oscillated between 71% and 84%. These values are similar to those reported in snacks made with potato flakes, grits, and starch, at a ratio of 25:25:50 expanded by microwave (74.50 ± 1.49%) [56].

The most porous sample was the expanded Tt snack, and the least porous was the expanded OtW snack. This may be related to the low protein content of OtW root (1.63% ± 0.07%) [25,57] compared to Tt (9.12% ± 0.13%) [25,52]. As previously discussed (Section 3.2), this component is essential because low values produced extrudates with a dense structure and small average cell size (Figure 6). However, the bulk density and porosity did not show significant differences (*p <* 0.05) between the evaluated samples (Table 2).

#### 3.3.3. Hygroscopicity

Hy varied significantly (*p <* 0.05) between the dried 3G pellets and the expanded snacks (Table 2). Expanded snacks have greater Hy, which can pose a challenge during packaging and marketing because the product has the potential to absorb moisture [58]. Therefore, Hy indicates quality control because small moisture changes can diminish the snack’s crispiness [59]. Expanded snacks made with OtY have high Hy values. According to scientific evidence, this could be because these samples have numerous pores with thin walls and an open-pore structure [60]. In contrast, the control sample is the least hygroscopic. This trend is related to the low values of WSI and SWE observed in this sample. This could indicate a lower presence of hydrophilic molecules in this sample and, therefore, a more significant presence of high molecular weight molecules that are hydrophobic, such as proteins [61]. According to García-Segovia [32], the Hy values in extruded snacks enriched with insects (*Alphitobius diaperinus* and *Tenebrio molitor*) and pea protein ranged from 30 to 33 g_w_/100 g_dry solid_. However, values of 25.5 to 26.8 g_w_/100 g_dry solid_ were reported for corn-extruded snacks with the addition of *Urtica dioica* [62]. This shows that the snacks developed in this study have lower values compared to others of the same species, mainly due to a relationship between this property and protein content.

#### 3.3.4. Water Absorption Index (WAI), Water Solubility Index (WSI), and Swelling Index (SWE) of Expanded Snacks

The WAI index shows significant differences (*p <* 0.05) between the control and the rest of the expanded snacks (Table 2). This index is relevant because it is an indicator of starch gelatinization. The higher the gelatinization degree, the greater the water absorption [63]. The WSI index represents the reduction in the molecular weight of pectin and hemicellulose molecules during extrusion. A higher WSI value is observed in OtR (4.33 ± 0.48). Therefore, this expanded snack suffered significant damage to the water-solubilized molecules. The lowest value was observed in the control (2.06 ± 0.25). These results suggest that WAI and WSI are related to the level of extrudate expansion, because this process produces starch gelatinization and breakdown of pectin and hemicellulose.

In contrast to what was observed in corn snacks made with a partial replacement of broccoli flour (5%) and alfalfa flour (5%) [64], in this study, the WAI index decreased and the WSI index increased with the addition (20%) of tuber flour. This behavior may be due to the tuber’s fiber content influences in these characteristics, since there is scientific evidence that the increase in fiber (from 0 to 20%) causes a decrease in the WAI index and an increase in the WSI index [65].

Starch gelatinization, protein denaturation, and non-starch polysaccharides fragmentation are the most critical reactions during extrusion. The ability to swell in water reflects the interaction between the native starch granules to adsorb water and release the starch, which results in a gelatinization process when cooled. The SWE index is higher in OtW and lower in the control.

#### 3.3.5. Texture Properties of Expanded Snacks

A puncture test was performed to simulate the first bite in the expanded snacks. Crispness work (W_c_), average specific force of structural ruptures (Fs), and average puncturing force (F_p_) showed higher values for the C and OtR expanded snacks (Table 3). The high value of W_c_ shows that these samples are harder than the rest. The lower W_c_ value was presented in Tt, demonstrating that more porous snacks exhibited a lower hardness (discussed in Section 3.3.2). Furthermore, the expanded Tt snack showed values lower than the other samples for F_s_ and F_p_. This indicates that consumers will perceive these snacks as easier to chew because they require less force to crush. The spatial frequency of structural ruptures (N_sr_) did not present differences between the samples. N_sr_ describes the number of fracture events that occur during compression. Finally, IbP presented more peaks (N_0_) during the analyses (61 ± 4), and the OtY sample had fewer peaks (50 ± 6). This parameter is related to the force required to break the polymer membranes of gas inclusions.

A correlation analysis was applied to explain the relationship between W_c_, F_s_, F_p_, and ρ_b_, WAI, and Hy. The W_c_, F_s_, and F_p_ were positively correlated with WAI (0.7070, 0.7371, and 0.6180, respectively, *p <* 0.05), as previously reported in quinoa snacks [66], yam-corn-rice snacks [67], and corn-Medicago sativa snacks [68]. Furthermore, W_c_ and F_s_ were negatively correlated with ρ_b_ and Hy (−0.7321 and −0.6620, *p <* 0.05; −0.6110 and −0.6355, *p <* 0.05, respectively). Therefore, the higher the bulk density and Hy of the expanded snacks, the higher the crispness work and the force of structural ruptures. The relationship between textural properties and Hy should be studied more extensively, because the crispness of this type of product can be affected when hydrated. When a snack exceeds a critical water activity value of 0.5, it is considered to lose its characteristic textural attributes [69].

#### 3.3.6. Optical Properties of Mixture, Dried 3G Pellets, and Expanded Snacks

Numerous reactions occur during extrusion that affects the color of the snack. The most common are non-enzymatic browning reactions (Maillard reaction and sugar caramelization) and pigment degradation [70]. Analysis of the influence of the samples on optical properties showed that the luminosity, chroma, and hue values differ significantly (*p <* 0.05) between the samples (Table 4). Expanded snacks made with IbP have lower chroma and hue values in all phases (mixing, extrusion, and expansion). In contrast, control expanded snacks showed higher luminosity, chroma, and hue values in most processes.

Analysis of the influence of the process on optical properties showed that extrusion and drying caused a decrease in luminosity values, with the IbP, OtR, and Tt samples showing the most significant reduction (ΔL* of −32.29, −31.10, and −30.65, respectively). While the expansion (microwave drying) process showed an increase in luminosity in all samples (compared between the dried 3G pellet and the expanded snack), the OtW and OtR snacks showed the most significant variation (ΔL* of 27.14 and 25.44, respectively).

The chroma (C*) analysis showed two types of behaviors. In the first case, the samples reflect an increase in the chroma values in the extrusion process and a decrease after the microwave-driven expansion. This conduct was shown by snacks made with IbP, OtW, OtY, and OtR. This trend is expected because the extrusion process generates non-enzymatic browning reactions and pigment degradation [71]. Furthermore, this process reduces water, producing closed (less porous) structures, which visually causes saturated colors (Figure 7). However, after microwave-driven expansion, the structure of the expanded snacks is more porous, which visually denotes less intense colors, as long as the process is not too intense and generates the appearance of undesirable colors.

The second behavior shows an opposite trend (chroma decrease in the extrusion process and increase in microwave-driven expansion), with C and Tt snacks showing this trend. These results are related to the soluble sugar content of corn grits and mashua (4 and 7 °Brix, respectively). During cooking extrusion, the reducing sugars (glucose and fructose) react with the amino acids present in the sample by Maillard and caramelization reactions, creating brown pigments [72]. This indicates that the higher the °Brix in the mixture, the more likely the pellets will generate brown colors. Furthermore, this would explain the more significant variation in chroma observed in the IbP, OtW, OtY, and OtR samples, tubers, and roots that have Brix values of 12 ± 1.3°.

The determination of the global color variation (ΔE_1_ and ΔE_2_) showed higher values in the control sample. However, the sample with the least color variation was Tt. This may be related to the chemical and nutritional changes that occur in the sample during the different processes, and generates a hypothesis that can be verified in future studies.

#### 3.3.7. Bioactive Compounds of Expanded Snacks

The results of the TC, lycopene (LP), TP, and AC of the expanded snacks are shown in Table 5. The main problems detected in comparing these data with other studies are the differences in how the data are expressed, the raw treatments before the analysis, and the geographical location.

Regarding TC, the values showed a significant variation, between 2.11–7.21 mg β-carotene/100 g of a sample. Its high AC characterizes Mashua (Tt). In this study, Tt had the highest carotenoid value (7.21 mg β-carotene/100 g of sample) (*p <* 0.05), with values above those reported by Campos [73] in 11 mashua genotypes (0.1–2.5 mg β-carotene/100 g of sample). The lowest value was detected in IbP, probably due to the anthocyanin content characteristic of purple camote. However, the OtW, OtY and OtR showed values of 2.12–3.11 mg β-carotene/100 g, similar to those reported by Campos in 14 fresh Oca genotypes [73]. Likewise, Tt snacks showed the highest value of lycopene content (*p <* 0.05); in the other crops, there was no significant difference (*p* > 0.05). Tomatoes present a high content of lycopene (8.8 to 42.0 μg/g_wet basis_), which is an active biological phytochemical [74].

The total phenol content determined in Andean crops could be related to phenolic acids, tannins, and flavonoids. These components could also be influenced by the state of maturity, geographical location, and damage caused by the drying process [25]. The Folin–Ciocalteu assay is an extensive method for determining total phenol content. However, other substances, such as proteins and sugars, could react with the reagent. It is essential to consider this, because proteins and sugars are part of the root’s composition [75]. Values of approximately 110 to 129 mg GA/100 g were found in most developed snacks, and the lowest phenol content corresponded to the control (*p <* 0.05). The phenol content in most of the extruded snacks with Andean crops is similar to those reported by Catunta [76] in fresh samples of mashua (128–146 mg GA/100 g) by Pacheco [77], who reported 100 mg GA/100 g, and by Chirinos [2], who reported a value of 2.32 mM trolox/100 g_dw_ in mashua cultivated in Junín (Peru). The phenolic content is essential due to its great health benefits when consumed sufficiently to generate a positive effect. Phenolic compounds protect cells from free radicals, reduce the risk of cell deterioration due to age, and improve immune system functions [78].

The highest AC was found in the Tt snack; a slight difference was observed in the IbP snack (*p <* 0.05). In contrast, the Ot snacks showed the lowest AC, with no differentiation between the varieties (OtW, OtY, and OtR) (*p* < 0.05). The AC can be attributed to compounds such as flavonoids and other phenolic compounds, as well as carotenes and vitamin C, which are present in these roots [73,79].

Correlational statistical analyses were performed to explain the influence of TC, LP, and TP on the AC of the samples. TP played a significant role in AC (0.9343, *p* < 0.05), followed by LP (0.7240, *p* < 0.05) and TC (0.6424, *p* < 0.05).

## 4. Conclusions

The dehydration curves of dried 3G pellets were better adjusted with the Page model; however, they also showed acceptable fits in the logarithmic and Midilli–Kucuk models. The Page model’s constant (k) showed a higher value in OtR and C, indicating a higher water diffusion velocity in these samples. According to the SEI values and appearance, the best treatment is with the IbP snack with a microwave application time of 70 s. The interaction of starch and protein in this sample may influence the increase in the paste viscosity during the expansion process. This should be verified in subsequent studies by analyzing protein content, protein denaturation, and past viscosity. The influence of the raw material’s water content and water activity was observed during the characterization in the different processing phases. However, the results of WAI and WSI suggest that thermal degradation, depolymerization, and recombination of fragments occur during extrusion. Textural analysis showed a relationship between porosity and hardness. The change in the optical properties could have been caused by the loss of the reducing sugars (glucose and fructose) during Maillard reactions with proteins. The optical analysis and AC showed that the Tt sample suffered fewer chemical changes or nutritional loss during the process. Finally, the extrusion process showed interesting results for manufacturing snacks from Andean tuber flours. Based on the results obtained in this study, the authors highlight the unique properties of these crops with the objective of promoting the cultivation, consumption, and development of products within and outside the Andean region.

## Figures and Tables

**Figure 1 foods-12-02168-f001:**
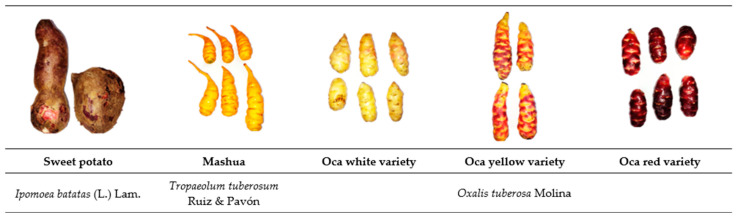
Andean tubers and tuberous roots.

**Figure 2 foods-12-02168-f002:**
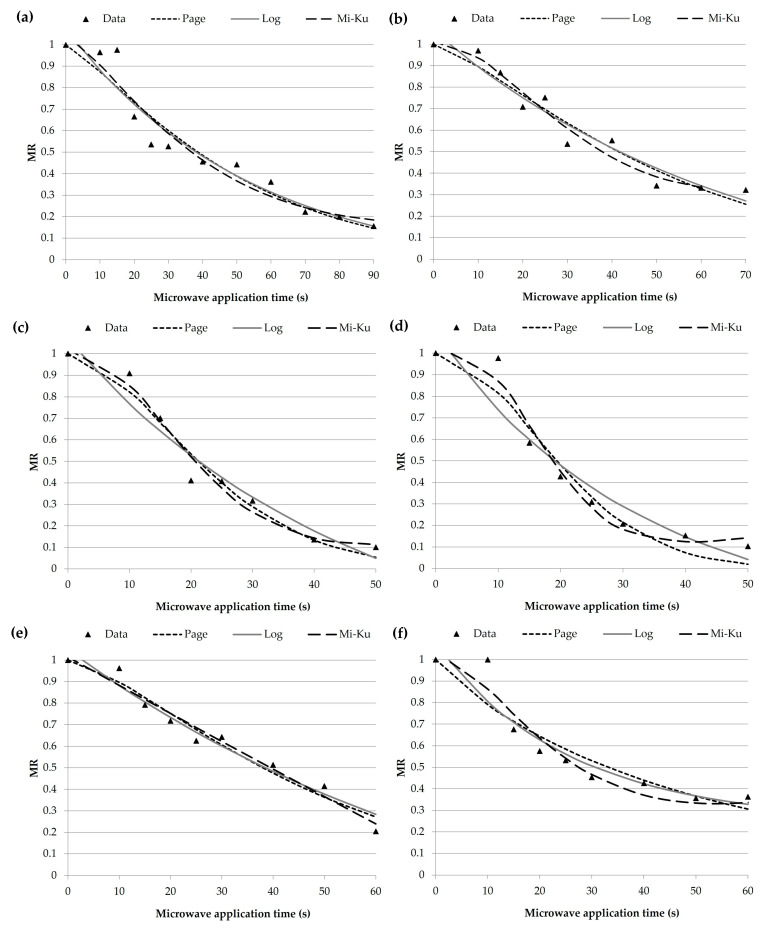
Microwave dehydration kinetics of dried 3G pellets made with corn and Andean flour in an 80:20 ratio (80% corn grits: 20% sample flour) with 25% wb of water (MR represents the amount of moisture remaining in the samples reported in relation to the initial moisture content). (**a**) C: Control snack, (**b**) IbP: Purple sweet potato snack, (**c**) Tt: Mashua snack, (**d**) OtW: Oca white variety snack, (**e**) OtY: Oca yellow variety snack, and (**f**) OtR: Oca red variety snack; adjusted to Page, logarithmic (log), and Midilli–Kucuk (Mi–Ku) models.

**Figure 3 foods-12-02168-f003:**
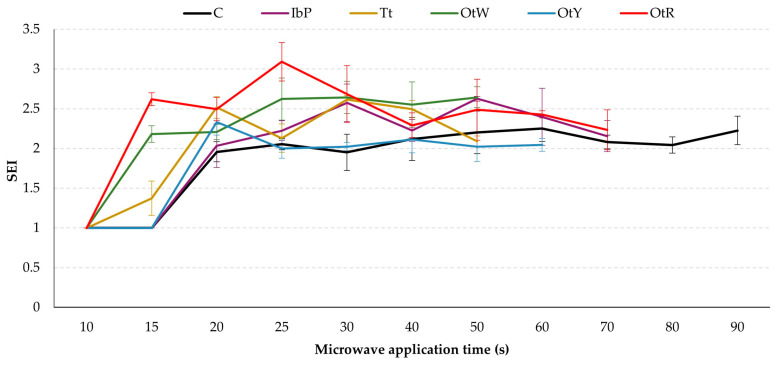
SEI evolution of the dried 3G pellets as a function of the microwave drying time (C: Control, IbP: purple sweet potato, Tt: mashua, OtW: Oca white variety, OtY: Oca yellow variety, OtR: Oca red variety).

**Figure 4 foods-12-02168-f004:**
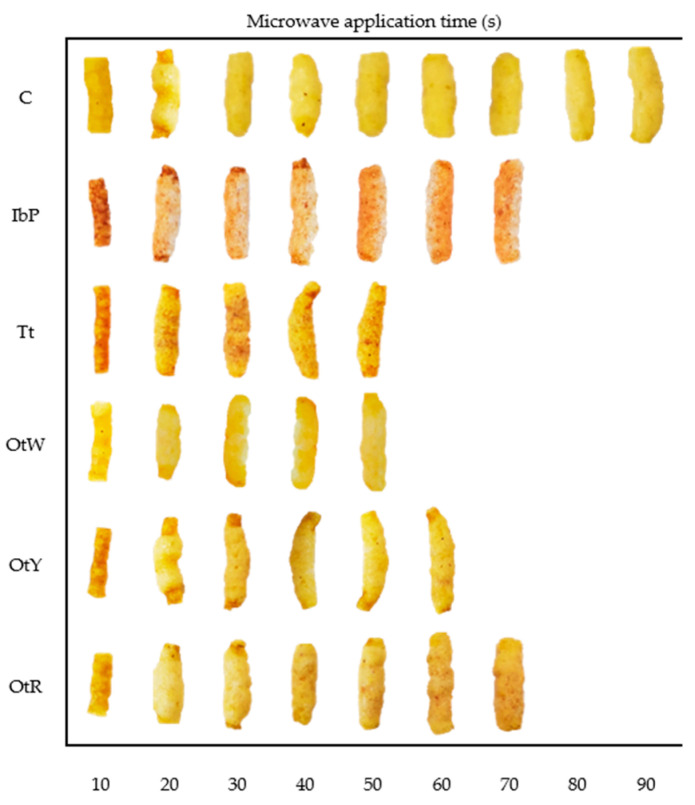
Appearance evolution of dried 3G pellets as a function of microwave drying time. (C: Control, IbP: purple sweet potato, Tt: mashua, OtW: Oca white variety, OtY: Oca yellow variety, OtR: Oca red variety).

**Figure 5 foods-12-02168-f005:**
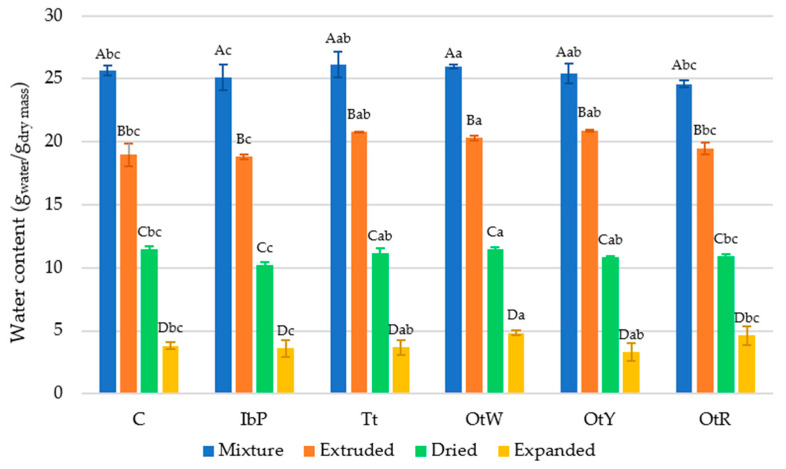
Mean values and standard deviations of the water content of mixture, extruded 3G pellets, dried 3G pellets, and expanded snacks. Samples, C: Control, IbP: purple sweet potato, Tt: mashua, OtW: Oca white variety, OtY: Oca yellow variety, OtR: Oca red variety. Different capital letters represent significant differences (*p <* 0.05) by process, and lowercase letters represent significant differences (*p <* 0.05) by samples.

**Figure 6 foods-12-02168-f006:**
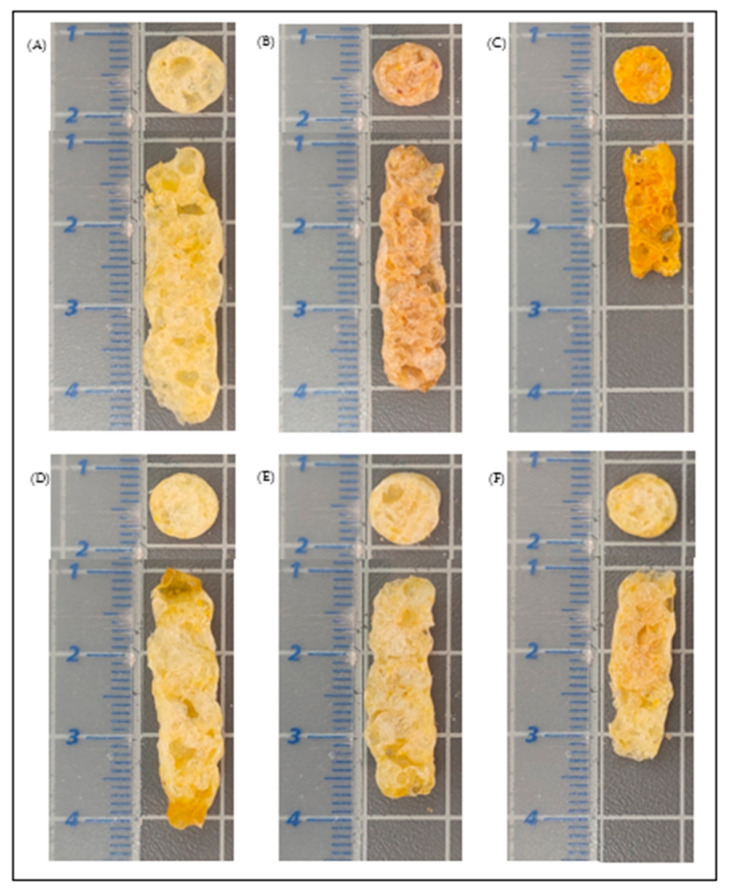
Transversal and longitudinal section of third-generation snacks. (**A**): Control (C); (**B**): Purple sweet potato (IbP); (**C**): Mashua (Tt); (**D**): Oca white variety (OtW); (**E**): Oca yellow variety (OtY); (**F**): Oca red variety (OtR).

**Figure 7 foods-12-02168-f007:**
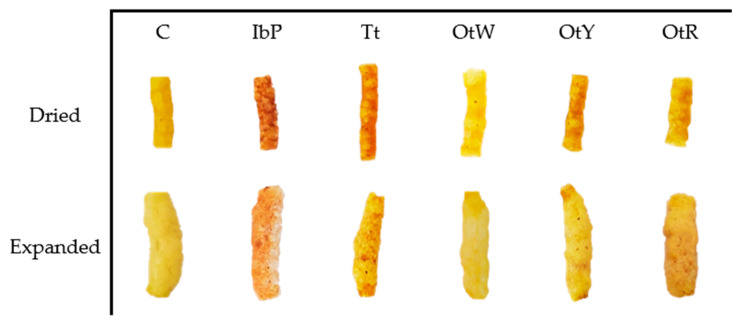
The appearance of pellets after extrusion and products expanded by microwaves.

**Table 1 foods-12-02168-t001:** Parameters obtained in the microwave drying kinetic of dried 3G pellets (C: Control, IbP: purple sweet potato, Tt: mashua, OtW: Oca white variety, OtY: Oca yellow variety, OtR: Oca red variety) adjusted to Page, logarithmic, and Midilli–Kucuk models.

Sample	Model			
		Page	Logarithmic	Midilli–Kucuk
C	Model constants	k: 0.00833 n: 1.2098	a: 1.1289 k: 0.0186 c: −0.0561	a: 1.0476 k: 0.0086 n: 1.2679 b: 0.0012
	Adj. R^2^	92.19	91.4	91.09
	RMSE	0.085	0.089	0.091
IbP	Model constants	k: 0.00562 n: 1.29293	a: 1.2525 k: 0.0142 c: −0.1926	a: 1.0131 k: 0.0029 n: 1.6223 b: 0.0037
	Adj. R^2^	93.83	92.31	94.37
	RMSE	0.065	0.072	0.062
Tt	Model constants	k: 0.00408 n: 1.68135	a: 1.5536 k: 0.0212 c: −0.4891	a: 1.0221 k: 0.0028 n: 1.8694 b: 0.0019
	Adj. R^2^	96.19	91.93	95.61
	RMSE	0.066	0.096	0.071
OtW	Model constants	k: 0.00297 n: 1.83746	a: 1.3471 k: 0.03 c: −0.2597	a: 1.0371 k: 0.0013 n: 2.2165 b: 0.0028
	Adj. R^2^	93.76	86.93	95.38
	RMSE	0.089	0.128	0.076
OtY	Model constants	k: 0.00454 n: 1.38244	a: 1.5418 k: 0.0114 c: −0.4939	a: 1.0141 k: 0.0032 n: 0.9881 b: −0.0101
	Adj. R^2^	95.67	94.51	94.83
	RMSE	0.053	0.06	0.058
OtR	Model constants	k: 0.02928 n: 0.903809	a: 0.8212 k: 0.0381 c: 0.2443	a: 1.0363 k: 0.0088 n: 1.4406 b: 0.0049
	Adj. R^2^	87.21	88.56	90.62
	RMSE	0.09	0.085	0.077

a, b, c, and k: Drying constants. n: Drying exponent. Adjusted regression coefficient (Adj. R^2^) and root-mean-square error (RMSE) values.

**Table 2 foods-12-02168-t002:** Mean values (and standard deviations) of the dried 3G pellets (water activity (a_w_^P^) and hygroscopicity (Hy^P^)), and expanded snacks (hygroscopicity (Hy^E^), bulk density (ρ_b_), porosity (ε) water absorption index (WAI), water solubility index (WSI), and swelling index (SWE)).

Sample	aw^P^	Hy^P^	Hy^E^	ρ_b_	ε	WAI	WSI	SWE
		(g_w_/100 g_dry solid_)	(g_w_/100 g_dry solid_)	(g/cm^3^)	%		%	(mL_swollen_/g_dry solid_)
C	0.303 (0.003) ^d^	4.05 (0.03) ^Bc^	9.80 (0.49) ^Ac^	0.15 (0.0005) ^a^	77.34 (1.92) ^a^	6.18 (0.12) ^a^	2.06 (0.25) ^b^	5.00 (0.26) ^d^
IbP	0.233 (0.003) ^e^	6.27 (0.35) ^Bab^	15.81 (1.23) ^Aab^	0.20 (0.03) ^a^	73.35 (3.91) ^a^	5.34 (0.26) ^b^	4.04 (0.74) ^ab^	6.69 (0.29) ^abc^
Tt	0.348 (0.003) ^b^	6.57 (0.14) ^Bab^	13.22 (0.72) ^Aab^	0.21 (0.03) ^a^	84.16 (2.13) ^a^	4.69 (0.31) ^b^	3.86 (0.78) ^ab^	5.37 (0.06) ^cd^
OtW	0.314 (0.003) ^c^	5.20 (0.05) ^Bab^	14.35 (1.34) ^Aab^	0.23 (0.01) ^a^	71.04 (0.65) ^a^	5.19 (0.11) ^b^	4.05 (0.23) ^ab^	8.04 (0.59) ^a^
OtY	0.37 (0.003) ^a^	5.96 (0.12) ^Ba^	16.16 (1.99) ^Aa^	0.23 (0.04) ^a^	72.19 (5.60) ^a^	5.02 (0.11) ^b^	4.08 (0.27) ^ab^	6.29 (0.10) ^bcd^
OtR	0.315 (0.003) ^c^	6.10 (0.08) ^Bb^	12.92 (0.20) ^Ab^	0.18 (0.02) ^a^	74.79 (5.16) ^a^	5.22 (0.18) ^b^	4.33 (0.48) ^a^	7.46 (0.53) ^ab^

Different capital letters represent significant differences (*p <* 0.05) by process (P: dried 3G pellet, E: expanded snacks), and lowercase letters represent significant differences (*p <* 0.05) by samples (C: Control, IbP: purple sweet potato, Tt: mashua, OtW: Oca white variety, OtY: Oca yellow variety, OtR: Oca red variety).

**Table 3 foods-12-02168-t003:** The texture parameters (mean values and standard deviations) of expanded snacks: crispness work (W_c_), average specific force of structural ruptures (Fs), average puncturing force (F_p_), spatial frequency of structural ruptures (N_sr_), and number of peaks (N_0_).

Sample	W_c_	F_s_	F_p_	N_sr_	N_0_
	(N × mm)	(N)	(N)	(mm^−1^)	
C	0.82 (0.07) ^a^	6.0 (0.8) ^a^	5.1 (0.3) ^a^	7.3 (0.7) ^a^	57 (6) ^abc^
IbP	0.57 (0.12) ^b^	4.3 (0.8) ^b^	3.6 (0.9) ^b^	7.8 (0.8) ^a^	61 (4) ^a^
Tt	0.41 (0.09) ^b^	3.1 (0.6) ^c^	2.4 (0.4) ^c^	7.5 (0.7) ^a^	51 (7) ^bc^
OtW	0.46 (0.08) ^b^	3.9 (0.9) ^bc^	3.4 (0.9) ^b^	7.7 (0.6) ^a^	59 (4) ^ab^
OtY	0.50 (0.03) ^b^	3.9 (0.7) ^bc^	3.3 (0.6) ^bc^	7.5 (0,8) ^a^	50 (6) ^c^
OtR	0.78 (0.13) ^a^	5.6 (0.7) ^a^	4.9 (0.7) ^a^	7.4 (0.9) ^a^	54 (8) ^abc^

Lowercase letters represent significant differences (*p <* 0.05) by samples (C: Control, IbP: purple sweet potato, Tt: mashua, OtW: Oca white variety, OtY: Oca yellow variety, OtR: Oca red variety).

**Table 4 foods-12-02168-t004:** Color coordinates (L*, a*, b*, C* y h*) (mean values and standard deviations), total color differences between the mixture before extrusion and the finished expanded product (ΔE_1_), and total color differences between the dried 3G pellet and the finished expanded snack (ΔE_2_).

Sample	L*	a*	b*	C*	h*	ΔE_1_	ΔE_2_
Mixtures before extrusion							
Control	77.93 (0.32) ^Aa^	6.72 (0.70) ^Cd^	42.38 (2.09) ^Bc^	42.91 (2.17) ^Bc^	81.01 (0.51) ^Bc^		
IbP	73.45 (1.02) ^Ae^	4.61 (0.26) ^Cb^	19.05 (1.58) ^Be^	19.59 (1.60) ^Be^	76.39 (0.36) ^Bf^		
Tt	69.27 (1.31) ^Af^	9.93 (1.27) ^Ca^	43.03 (4.75) ^Ba^	44.16 (4.92) ^Ba^	77.03 (0.22) ^Be^		
OtW	74.39 (2.72) ^Ac^	6.01 (0.58) ^Cc^	34.85 (5.43) ^Bb^	35.36 (5.45) ^Bb^	80.16 (0.56) ^Bb^		
OtY	76.39 (0.21) ^Ad^	6.45 (0.05) ^Cb^	41.11 (0.04) ^Bab^	41.61 (0.05) ^Bb^	81.09 (0.06) ^Bd^		
OtR	80.63 (1.15) ^Ab^	4.29 (0.30) ^Ce^	26.17 (2.23) ^Bd^	26.52 (2.25) ^Bd^	80.68 (0.14) ^Ba^		
Dried 3G pellets							
Control	65.39 (1.18) ^Ca^	7.51 (0.49) ^Ad^	25.56 (2.14) ^Bc^	26.64 (2.19) ^Bc^	73.60 (0.33) ^Cc^		
IbP	41.16 (0.26) ^Ce^	10.57 (0.08) _Ab_	25.87 (0.23) ^Be^	27.94 (0.25) ^Be^	67.77 (0.03) ^Cf^		
Tt	38.62 (0.00) ^Cf^	13.59 (0.01) _Aa_	32.67 (0.00) ^Ba^	35.38 (0.00) ^Ba^	67.42 (0.01) ^Ce^		
OtW	49.39 (0.10) ^Cc^	9.74 (0.005) _Ac_	41.59 (0.22) ^Bb^	42.71 (0.21) ^Bb^	76.82 (0.07) ^Cb^		
OtY	48.57 (0.17) ^Cd^	11.14 (0.01) _Ab_	38.46 (0.15) ^Bab^	40.04 (0.14) ^Bb^	73.85 (0.07) ^Cd^		
OtR	49.53 (0.00) ^Cb^	7.90 (0.01) ^Ae^	38.28 (0.005) ^Bd^	39.09 (0.004) ^Bd^	78.34 (0.009) ^Ca^		
Expanded snacks							
Control	79.56 (0.09) ^Ba^	3.89 (0.04) ^Bd^	37.35 (0.25) ^Ac^	37.55 (0.25) ^Ac^	84.06 (0.02) ^Ac^	73.06 (0.58) ^a^	74.40 (0.68) ^a^
IbP	63.77 (0.02) ^Be^	7.87 (0.005) ^Bb^	24.84 (0.01) ^Ae^	26.06 (0.01) ^Ae^	72.43 (0.02) ^Af^	60.25 (0.58) ^d^	57.82 (0.17) ^d^
Tt	63.14 (0.005) ^Bf^	14.00 (0.005) ^Ba^	49.36 (0.005) ^Aa^	51.31 (0.01) ^Aa^	74.17 (0.005) ^Ae^	54.10 (1.95) ^e^	57.76 (0.005) ^d^
OtW	76.52 (0.01) ^Bc^	4.43 (0.01) ^Bc^	38.19 (0.01) ^Ab^	38.44 (0.01) ^Ab^	83.39 (0.01) ^Ab^	70.82 (0.66) ^b^	72.17 (0.03) ^b^
OtY	71.48 (0.02) ^Bd^	5.89 (0.005) ^Bb^	39.20 (0.01) ^Aab^	39.63 (0.01) ^Ab^	81.46 (0.01) ^Ad^	65.25 (0.06) ^c^	64.55 (0.06) ^c^
OtR	74.97 (0.03) ^Bb^	3.77 (0.02) ^Be^	31.99 (0.05) ^Ad^	32.21 (0.06) ^Ad^	83.27 (0.02) ^Aa^	71.17 (0.59) ^ab^	72.00 (0.005) ^b^

Different capital letters represent significant differences (*p <* 0.05) by process (mixture before extrusion, dried 3G pellet, expanded snack), and lowercase letters represent significant differences (*p <* 0.05) by samples (C: Control, IbP: purple sweet potato, Tt: mashua, OtW: Oca white variety, OtY: Oca yellow variety, OtR: Oca red variety).

**Table 5 foods-12-02168-t005:** Total carotenoids (TC), lycopene (LP), total phenols (TP) and antioxidant capacity (AC) (mean values and standard deviations) of expanded snacks.

Sample	TC	LP	TP	AC
	(mg_β-carotene_/100 g)	(mg/100 g)	(mg_GA_/100 g)	(mg_Trolox_/100 g)
C	2.00 (0.02) ^d^	0.974 (0.007) ^b^	110.9 (1.3) ^b^	2.4 (0.2) ^d^
IbP	2.11 (0.03) ^cd^	1.07 (0.03) ^b^	128 (2) ^a^	8.18 (0.09) ^b^
Tt	7.21 (0.08) ^a^	1.68 (0.06) ^a^	129 (3) ^a^	9.3 (1.2) ^a^
OtW	2.18 (0.06) ^c^	1.02 (0.05) ^b^	115.9 (1.2) ^b^	5.5 (0.3) ^c^
OtY	3.11 (0.09) ^b^	1.04 (0.07) ^b^	116 (3) ^b^	5.38 (0.05) ^c^
OtR	2.12 (0.05) ^cd^	0.995 (0.004) ^b^	117 (2) ^b^	5.78 (0.09) ^c^

Different lowercase letters in the columns represent significant differences (*p <* 0.05) by samples (C: Control, IbP: purple sweet potato, Tt: mashua, OtW: Oca white variety, OtY: Oca yellow variety, OtR: Oca red variety).

## Data Availability

Data is contained within the article.

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
