# Peer review of "Third-Generation Snacks Manufactured from Andean Tubers and Tuberous Root Flours: Microwave Expansion Kinetics and Characterization"

_foods, 2023, doi:10.3390/foods12112168_

Round 1
Reviewer 1 Report
The manuscript of Acurio, Salazar, Garcia-Segovia, Martinez-Monzo and Igual reports on the dehydration kinetics and MW expansion of blends of corn and tuberous root flours. Pellets were produced by low moisture extrusion. Physical and dimensional properties were studied, plus the antioxidant capacity of snacks. The described unit operations are blending, extrusion, first drying at RT, and MW dehydration-expansion, focusing on the last operation.
The manuscript requires careful revision to clarify different points and prevent any misunderstanding. Ideally, the authors should discuss more substantial links to compositional aspects (starch, protein, fibre, low molecular weight carbohydrates) to product structure and texture to improve the scientific value. The discussion of varying drying kinetics and expansion must be better aligned and justified.
For drying, kinetics were studied, and three different models were applied. The time dependency was registered for the expansion, but modelling is not included. This leads to confusion because the authors use "expansion kinetics" to describe the work.
L98: How did the authors obtain from dried slices the flour?
L114: Information on the relative humidity of air and air circulation or not is required. Does the final moisture represent the equilibrium moisture or still depend on the initial moisture, which differs slightly between the samples?
L207: The MM section must present equations for the different models.
L208: Abbreviations must be introduced for fluent reading. Even in 2.1, abbreviations are not mentioned, but in Tab. 1 only. Improve readability, please.
L217: In l 220 (k) is called a drying constant. Why (k) is mentioned here to be a diffusion coefficient? Do the authors mean pore diffusion in the expanded product, the starchy layer, or all-over diffusion through the porous material with different solid and gas layers?
L223-225: Do the authors provide data about such phenomena, or is this from literature? Please clarify.
L227: Please mention the meaning of MR in the caption and prevent conflicts with the term "expansion kinetics", which suggests information upon SEI or LEI. Here it would be instead "dehydration kinetics" according to subtitle 3.1. / Mention also the initial moisture of extrudates in the title so that the figure is self-explaining. / How many measurements per data point are available (average, sd, error bars)?
L231: The MR might correlate with expansion, but this still needs to be clarified at this stage of developing the idea. Fit parameters are for MR, most probably (dehydration kinetics). / The comparison of fit parameters with the current layout of the table could be more convenient. E.g., in the text, R^2 is compared for different models and materials, so finding values in the table might be improved somehow.
L248: Could the authors be more explicit - as lower the starch content, as lower the past viscosity, as higher the SEI? Is this an effect of starch content (see lines 249-251) or protein content (see lines 247-249)? / Is the composition of OtY significantly different from OtW and OtR? / Could the authors also provide information about the amylose: amylopectin ratio of the different starches?
L260: The reduction is not 22% for the extrusion process but for all unit operations. Please rephrase and clarify this point.
L268: What exactly is meant: "Stable moisture" could be equilibrium water content at a specific relative humidity. In this case, a sorption-desorption isotherm would be helpful.
L283: See comment in tab. 2.
L292: This is not a surprise as such. However, is water activity a function of the composition of biomolecules or random flour moisture?
L317: That products absorb potentially water is very generic. More important is to highlight the relationship between water content and crispiness. Did the authors detect crispiness as a function of moisture content?
L320: Is it specific and measured that expanded OtY samples have open pores, or is this true for all expanded products? For explaining results, do not use hypothetical properties.
L321: "least hygroscopic"... because of a lower porosity or protein content? Did the observed Hy correlate with porosity (larger specific surface) or protein content? Try to discuss such points to find the most probable conclusion. The conclusion in line 326 is a simple classification but needs to be better linked to structure or composition.
L337: This paragraph is confusing and must be clarified. Different hypothesis conflict: WAI à degree of starch gelatinization, WSI à breakdown of pectin and hemicellulose, WAI and WSI - expansion, or gelatinization and starch breakdown are linked.
L342: The sentence needs to be clarified and must be rewritten.
L346: The sentence does not make sense in the current form and must be rewritten.
L351: Delete the sentence in lines 351-352. The positive correlation between water content and SWE was highlighted before.
L353: The final water content was reported to be linked to the initial water content of the blends. Swelling is the ability of water absorption by an elastic matter, while a non-elastic material could absorb water but would not swell. Could the authors test the correlation of SWE and starch: denatured protein content or similar relations between composition and physical extrudate properties? This would be more relevant for the discussion than correlation tests between similar physical descriptors (e.g., water content, WAI, SWE).
L355: The samples should be in equilibrium with the air moisture for improved comparability. Does the a_w depend on the random initial moisture of the raw materials before blending?
L361: What is the "mastication of food process"? The first bite might be simulated with a puncture test, but not oral processing, including mixing with saliva.
L370: Is the correlation supposed or calculated?
L374: Cell walls or polymer membranes of gas inclusions (the extrudate represents a solid foam)?
L408: … expanded operation àproduct expansion or microwave-driven expansion.
L412: during expansion -> after expansion
L416: Extrusion process or MW expansion?
L418: heating extrusion -> cooking extrusion or high-temperature extrusion?
L418: Do the authors suggest corn and mashua contain sucrose? Or is it rather glucose and fructose?
L440: What exactly is the problem - comparing different categories such as AC, the origin of crops, and treatment times or temperatures? The idea needs to be clarified.
L444: Numbers of the same type of analysis must be rounded to the same digit. Precision is a property of the method and not an individual sample.
L455: Use the same names or abbreviations as in the tab. 5.
L485: Should it be "dehydration curves" instead of "microwave expansion"?
L488: k is the drying constant (see line 220). Diffusion of water and vapour, drying might combine phase transition and different transport processes. Do not merge diffusion and drying, which are other processes.
L491: This is possible because the protein might be denatured, but it must be expressed as a hypothesis because neither past viscosity nor denaturation nor protein content was measured.
L491: The influence of ... on...
L496: Corn contains rather starch, which might be degraded to glucose, followed by Maillard reactions. If the authors detected sucrose at relevant concentrations, they must support the hypothesis with data.
L500: Excellent is a non-scientific expression and is not recommended to classify results. The feasibility of extrusion cooking and MW expansion was demonstrated for tuber crops - corn blends.
In the majority, the quality of English is appropriate.
Author Response
Dear Reviewer, thank you very much for your valuable remarks and suggestions. We considered them highly appropriate, and we included all of them in the present version of the manuscript (changes in the new document were underlined in grey). We express our appreciation for your time and effort in the preparation of comments and very valuable suggestions. Thank you very much.

Reviewer 2 Report
The paper deals with an exciting topic for the food industry and presents valuable results. However, the paper requires improvement in writing and data presentation.
The paper needs rearrangement and rewriting of some sections in order to bring clarifications. While reading this material I had difficulties understanding what where actually the final products that were analyzed, because some characterizations were made for samples designated as extruded, dried, or expanded snacks like in Fig. 5, Table 4 while the rest of the results were related only to expanded 3G snacks. To avoid confusion, it would be useful to unify the terminology and somehow make it clear from which production stages the samples were taken ( for example on a process diagram). I.e. what did the authors mean by extruded snacks, dried snacks and expanded snacks? As I understand expanded snacks are the final product of interest (3G snacks) but are the former semi-final products?
Improve the discussion of data obtained by modelling dehydration kinetics. Why are they important and what is the practical relevance/significance of the findings?
Specific comments
Lines 71-78: This information is more for M&M section and not the introduction section.
Lines 424-427: The conclusion that Tt sample suffered low nutritional loss only by determining the change in global colour variation is insufficient and should be evidenced by performing proximate analyses of the samples.
Lines 499-500: Avoid using words like "excellent results in this study" and the conclusion that extrusion is "an ideal method" for manufacturing because this study was not that overwhelming and comparisons were not made with other production methods. Conclusions should be limited only to information extracted from processed data.
The weakness of the paper is that changes in starch were not the focus of the study and that the glycaemic index was not determined for the 3G snacks. Extruded and expanded products usually have high GI which is not always desirable.
Author Response

(The authors gave the same response as above.)

Round 2
Reviewer 1 Report
The manuscript improved considerably. Some minor questions are still open.
206 Change the expression MW expansion curves if it is a dehydration curve. It is a combined process, but when referring to Fig. 2, “expansion” is not a well-justified expression.
234 All of the drying models in this study are empirical and fitting parameters are not descriptors of underlying “natural” phenomena. What do you mean by “natural”? Please revise the sentence and clarify the idea.
343 I do not understand the classification “water-like molecules”. Is the comment about the molecular weight or the hydrophilicity of molecules?
368 How can starch interact with gelatinized “starch” granules? Most probably, the native starch granules swell and release the starch, which will gelatinize at cooling. So, what do the authors mean by the expression in l 368-369?
Author Response
We appreciate the reviewer’s helpful and insightful assistance with this manuscript.
